# Spontaneous and Piezo Polarization Versus Polar Surfaces: Fundamentals and Ab Initio Calculations

**DOI:** 10.3390/ma18071489

**Published:** 2025-03-26

**Authors:** Pawel Strak, Pawel Kempisty, Konrad Sakowski, Jacek Piechota, Izabella Grzegory, Eva Monroy, Agata Kaminska, Stanislaw Krukowski

**Affiliations:** 1Institute of High Pressure Physics, Polish Academy of Sciences, Sokolowska 29/37, 01-142 Warsaw, Polandkonrad@unipress.waw.pl (K.S.); izabella@unipress.waw.pl (I.G.); kaminska@ifpan.edu.pl (A.K.); 2Research Institute for Applied Mechanics, Kyushu University, Fukuoka 816-8580, Japan; 3Institute of Applied Mathematics and Mechanics, University of Warsaw, 02-097 Warsaw, Poland; 4CEA, PHELIQS, Grenoble INP, IRIG, University Grenoble-Alpes, 17 av. des Martyrs, 38000 Grenoble, France; eva.monroy@cea.fr; 5Institute of Physics, Polish Academy of Sciences, Aleja Lotnikow 32/46, 02-668 Warsaw, Poland; 6Faculty of Mathematics and Natural Sciences, School of Exact Sciences, Cardinal Stefan Wyszynski University, Dewajtis 5, 01-815 Warsaw, Poland

**Keywords:** spontaneous polarization, piezoelectricity, wurtzite, zinc blende, nitrides, BN, AlN, GaN, InN

## Abstract

In this study, the fundamental properties of spontaneous and piezo polarization and surface polarity were defined. It was demonstrated that the Landau definition of polarization as a dipole density could be used in infinite systems. Differences between bulk polarization and surface polarity were distinguished, thus creating a clear identification of both components. This identification is in agreement with numerous experimental data—red shift presence and absence for wurtzite and zinc blende multiquantum wells (MQWs), respectively. A local model of spontaneous polarization was created and used to calculate spontaneous polarization as electric dipole density. The proposed local model correctly predicted the c-axis spontaneous polarization values of nitride wurtzite semiconductors. In addition, the model’s results are in accordance with a polarization equal to zero for the zinc blende lattice. The spontaneous polarization values obtained for all wurtzite III nitrides are in basic agreement with earlier calculations using the Berry phase. Ab initio calculations of wurtzite nitride superlattices in Heyd–Scuseria–Ernzerhof (HSE) approximation were performed to derive polarization-induced fields in coherently strained lattices, showing good agreement with the polarization values. Strained superlattice data were used to determine the piezoelectric parameters of wurtzite nitrides, obtaining values that are in basic agreement with earlier data. Zinc blende superlattices were also modeled using ab initio HSE calculations, showing results that are in agreement with the absence of polarization in all nitrides in zinc blende symmetry.

## 1. Introduction

Polarization is an important macroscopic, vectorial quantity that emerges in systems of symmetry groups, allowing a system to attain nonzero values [1]. Spontaneous polarization is a specific aspect of this phenomenon, in which a system attains a state of nonzero polarization without any inference from the outside. This phenomenon is an inherent property of the system, and therefore, it is defined in isolation. On the other hand, it is relatively easy to affect the state of the system and induce polarization by the mere application of an electric field from the outside. The field breaks system symmetry, leading to polarization. Therefore, the determination of polarization, in general, and its spontaneous variation requires a precise definition of external conditions. Paradoxically, in some cases, the manipulation of the system from the outside supports the determination of spontaneous polarization, despite the fact that its definition assumes no such influence.

The most prominent group of standard semiconductors consists of those having wurtzite and zinc blende lattices. Despite noticeable/apparent similarities, in some respects, they are drastically different. Wurtzite crystalline symmetry allows for the occurrence of polarization, while zinc blende does not. Macroscopically, polarization occurs due to the relative shift in the center of a negative electron charge with respect to the position of the positive atomic core, i.e., the creation of electric dipole density [1]. This interpretation may also be applied to finite-sized systems, such as molecules or nano-objects [2].

Polarization effects are important; they affect the physical properties of semiconductor systems through the emergence of electric fields of various magnitudes and ranges. In large-sized systems, macroscopic electric fields are negligible due to charge screening, known as Debye–Hückel or Thomas–Fermi effects [3,4]. A much stronger influence of polarization-induced electric fields is observed in nanometer-scale systems. A clear positive example of polarization application is the localization of electrons by the electric field in GaN-based field-effect transistors (FETs) [5,6]. In laser diodes (LDs) and light-emitting diodes (LEDs), based on III-nitride multiquantum wells (MQWs), polarization-related electric fields are highly detrimental, reducing electron–hole wavefunction overlap and consequently radiative recombination efficiency by the so-called quantum-confined Stark effect (QCSE) [7,8,9,10,11]. In devices containing heterostructures, polarization difference entails a sheet charge and a surface dipole layer at the heterointerfaces [5,6,12].

Polarization was defined first by Nobel Prize winner L. D. Landau as electric dipole density, i.e., the magnitude of the electric dipole for a unit of volume or for a separated molecule [1]. While in the case of the molecule, its finite size does not cause any fundamental problems, infinite solid polarization as a bulk property was questioned. At the beginning, Martin claimed that a property cannot be obtained from unit cell calculation because of the charge transfer between various cells and contributions from the surface states [13]. Posternak et al. calculated the polarization of BeO, showing that spontaneous polarization was not accessible in the procedure using periodic boundary conditions (PBCs) [14]. This derivation was discussed later by Resta et al. [15]. Accordingly, Tagantsev claimed that there was no possibility of defining spontaneous polarization as a bulk property [16]. In a series of later papers, Springborg and Kirtman et al. showed that polarization as a bulk property is critically affected by edge termination, which cannot be removed by extending the size of the system to infinity [17,18,19,20,21]. They concluded that polarization as a bulk property cannot be uniquely determined, as the result depends on the boundaries and also on the shape of the simulated system [21]. This argument was also used by Spaldin, who showed that, depending on the termination, two different values of polarization could be obtained for different terminations within a simple Clausius–Mosotti model, in which continuous charge distribution is replaced by a set of positively and negatively charged ions [22]. In the case of continuous electron charge distribution, this translates into an infinite number of polarization values.

As a remedy, a different approach was developed in which the polarization change was calculated [23,24,25,26]. This idea was first proposed by King and Vanderbilt, who declared that polarization is equivalent to surface charge density, but the modulo of the charge unit is $e/A_{surf}$ or $2e/A_{surf}$ [23,24]. This is in agreement with the results obtained by Resta, who declared that polarization calculated as dipole density is an ill-defined quantity [25,26]. In general, this is a correct statement as the calculated dipole depends on the choice of the cell boundaries, which undergoes a jump when the displaced cell boundary is crossed by an atom. However, in agreement with Landau’s statement, Resta declared that polarization is an intrinsic bulk property. To resolve the problem, he proposed to calculate the polarization change expressed in terms of the geometrical phase or Berry phase related to the polarization current induced when the polarization of the system changes due to predefined transformation. It should be emphasized that Resta derived this expression from Landau’s definition [25]. In his derivation, Landau’s definition of polarization was transformed into a calculation of polarization current expressed in terms of Wannier functions [26]. The polarization change was determined as a bulk quantity that can be determined using a periodic unit cell. The property obtained was claimed by Resta to be the only valid polarization in bulk. Later, Resta extended this approach in the context of magnetization [27].

Using this definition, Fiorentini et al. calculated the polarization change for wurtzite AlN, GaN, and InN using the zinc blende lattice as a reference, i.e., assuming the zinc blende polarization is zero [28,29]. As expected, the spontaneous polarization in the zinc blende lattice should vanish due to symmetry. Therefore, the polarization change values obtained are considered to be the total spontaneous polarization of the wurtzite nitrides. Subsequently, Dreyer et al. employed the identical procedure. [30]. The difference was that the spontaneous polarization difference was calculated between wurtzite and the artificially designed hexagonal phase [30]. The latter has zero polarization due to mirror symmetry with respect to the xy plane. The results of Fiorentini et al. and Dreyer et al. are drastically different; the latter’s polarization values are more than one order of magnitude higher. Dreyer et al. claimed that this difference is due to the fact that polarization in zinc blende is not zero. Recently, Yoo et al. calculated the spontaneous polarization of wurtzite and zinc blende GaN and AlN [12]. They obtained wurtzite polarization values that are in agreement with the earlier results of Dreyer et al. In addition, nonzero-polarization values of zinc blende GaN and AlN were listed. The latter nonzero value could be asserted; nevertheless, the spontaneous polarization of the wurtzite should be independent from the reference value, provided that the correct procedure is used. If so, then from the obtained polarization difference, it follows that the polarization of the zinc blende is comparable to that of the wurtzite. This assertion is demonstrably false; consequently, the discrepancy must be elucidated.

In addition to the Berry phase direct approaches, the indirect route was utilized. This approach was based on the determination of electric fields from ab initio models employed in the modeling of polar quantum wells and polar surfaces. Ab initio calculations were used for simulations of the multiquantum wells (MQWs). These MQWs form active layers of light-emitting diodes (LEDs) and laser diodes (LDs) [9,10,11]. Naturally, as an example for calculation, AlGaN or GaInN solid solution-based wells and barriers are poor candidates; therefore, the polar simple GaN/AlN MQWs were considered. The properties of these structures are affected by the polarization-induced field along the 0z-axis. In such structures, which are embedded in the external solid, electric fields arise, due to the barrier–well polarization difference [8,9,31]. This is perfectly simulated by ab initio calculation of a single AlN/GaN period, as the total potential difference across the barrier–well structure is zero [8,9,10,11]. Such calculations have been carried out for the ideal wurtzite and zinc blende lattices, where Al and Ga atoms are located in the ideal lattices with either GaN or AlN lattice parameters [11]. The results proved that an electric field arises in the wurtzite lattice, but it is zero in the zinc blende lattice. Of course, these results are obtained within the precision of the potential averaging and finite system size, but it is estimated that the fields in the zinc blende are at least two orders of magnitude lower. These results do not prove that the zinc blende has zero polarization, only that the GaN and AlN polarizations are identical in the ideal zinc blende lattices strained either to GaN or to AlN. The relaxation of the lattice leads to the appearance of the fields, as the ideal zinc blende symmetry is broken by the strain. In summary, it is only a strong indication that the spontaneous polarization in a zinc blende is zero.

Another indirect approach was based on slab simulations used for surface modeling. Spontaneous polarization is defined as the polarization of the solid in the absence of an external electric charge [32]. As discussed by Boguslawski and Bernholz, this is equivalent to a zero electric displacement field throughout the system. This leads to the presence of the polarization-induced electric field in the sample. Application of the external field could compensate for this field, bringing it to zero [11]. The nitride slab, with no charged surface states at the specially formed boundaries, was subject to the external field to obtain zero fields inside. From the magnitude of the applied external field, the spontaneous polarization was deduced. However, the relation between the Berry phase and the slab results requires explanation. These simulations produced different values of polarization. These results require further verification as the slab contains fractionally charged surface states that could be additionally charged due to the external field, affecting the relation between the external field and the polarization. An attempt has recently been made to compare these differences [33]. The influence of the piezo effect was also investigated. Despite the large discrepancy in the magnitude of polarization, the differences are similar and could easily be compensated by the strain-induced piezo effects. Therefore, it was not possible to make a conclusive statement, the results being that all these value sets are possible. Thus, polarization in infinite solids has not been precisely determined.

A completely different status has been achieved in studies of the polarization of finite objects, i.e., molecules and nanoclusters [2]. The polarization of finite objects is defined as a total magnetic moment, which is calculated using various formulations and is also measured experimentally [34]. Electric dipole moments have been calculated and compared using a large number of different numerical methods [35]. Good accuracy has been achieved with errors in the order of a few percent. In this way, the problem of the polarization of finite objects has been solved.

The problem of spontaneous polarization in infinite solids is not yet solved. The purpose of this paper is devoted to resolving the difficulties and providing definitive answers to these questions. Therefore, we first define spontaneous polarization and discuss the difference between bulk polarization and the polar surface effects. Then, the new local calculation method of spontaneous polarization is presented. This will be described in Section 2, which is devoted to the basic model. As the current state of the field definitely requires a basic formulation, this section is introduced before the presentation of the calculation method. The results obtained for the nitrides BN, AlN, GaN, and InN are then presented. Both the local bulk model and the supercell data are discussed. Finally, the present results are critically compared with the data previously obtained.

## 2. The Model

Spontaneous polarization is a bulk property that gives rise to the electric field that affects the properties of a semiconductor. The emerging field in a flat uniform slab is independent of its thickness, i.e., it is equivalent to an electrical capacitor. Accordingly, the identical electric potential can be obtained assuming electric charge density on both polar surfaces. An identical type of contribution stems from the charged surface effect; therefore, these effects are intermingled.

The aforementioned scenario was used in the polarization theory by Spaldin [22]. In a tutorial approach, she demonstrated the application of the modern theory of polarization to experimental determination by the Sawyer–Tower method [22]. In particular, in the diagram in Figure 1 of Ref. [11], the author shows two representative unit cells that could lead to two opposite signs of polarization values. This is further confirmed by Figure 2 of Ref. [11], in which the author presents the two polarizations represented by the surface charges. This interpretation is compatible with the original arguments of King-Smith and Vanderbilt [23,24] and those of Ambacher et al. [36], who posit that the polarization could be represented by surface charge. It is easy to show that this picture leads to an oversimplified interpretation of the phenomenon.

To prove this, we follow the Spaldin argument as applied to wurtzite and zinc blende lattices. We applied a simple model to zinc blende GaN and wurtzite GaN slabs terminated by $GaN\left(11\pm 1\right)$ and $GaN\left(000\pm 1\right)$ polar surfaces as shown in Figure 1 and Figure 2, for zinc blende and wurtzite, respectively.

From the results obtained, it is clear that the polarization measured by the Sawyer–Tower method, as described in Ref. [22], gives nonzero results in both cases. On the other hand, the polarization in zinc blende must disappear due to symmetry requirements. The Spaldin implementation is therefore an oversimplified model of polarization phenomena in crystals. The misinterpreted results are due to the lack of separate treatment of the two different factors: (i) the polarization due to the electron shift in the bulk bonding and (ii) the surface effect due to the surface charge’s contribution. These are two different effects, which have been demonstrated experimentally. In the nano-object embedded in the large-sized zinc blende matrix, the electric field is zero, as shown by recombination measurements in GaAs/GaAlAs multiquantum wells, where the field-related red shift is absent. This is independent of the polarity of the external surfaces of the zinc blende matrix. In the case of wurtzite nitride nano-objects, such as nitride multiquantum wells, the red shift effect is observed. This result is the experimental difference between these two aspects of the polar systems [37].

When analyzing the first component, it is important to note that the polarization cannot be reduced to the electron transfer between atoms. In the case of the covalently bonded solid, polarization occurs due to charge transfer between the crystal and neighboring atomic states, the latter being understood as bonding states or, in ab initio terms, the valence states. Thus, the proposed scenario of the polarization emergence in the case of wurtzite and zinc blende lattices is presented in Figure 3. The polarization is due to bonding; i.e., the bonding tetrahedra should be counted. In the case of ionic compounds, this shift can be interpreted as a transition to other states, but it is necessary to distinguish between the surface effect and the bulk effect [38]. In Spaldin’s publication, these contributions are mixed. The correct interpretation is shown in Figure 3c. In this case, the polarization is zero; while using the Spaldin argument, it is not zero. This difference is due to the surface effect, which must be subtracted.

The set of isolated separate atoms has a polarization equal to zero, so the spontaneous polarization is equal to the polarization change achieved in the bonding. The polarization in mixed electron–proton form can be defined as [25](1a)$$\vec{P}=\frac{e}{V}\left(\int_{V}\vec{r}\rho_{tot}\left(\vec{r}\right)d^{3}r\right)=\frac{e}{V}\left(\sum_{j=1}^{N}Z_{j}{\vec{r}}_{j}-\int_{V}\vec{r}\rho_{el}\left(\vec{r}\right)d^{3}r\right)$$
where *j* represents the indices of all atoms, *N* is the number of atoms, and *V* is the volume. The mixed total charge density is(1b)$$\rho_{tot}\left(\vec{r}\right)=\sum_{j=1}^{N}Q_{j}\delta \left(\vec{r}-{\vec{r}}_{j}\right)-e\rho_{el}\left(\vec{r}\right)$$
where the charge of the nucleus of *j*-th atom is $Q_{j}=Z_{j}e$ and *e* is the elementary charge and the electron density, obtained from the summation over all basis functions, $\varphi_{q}\left(\vec{r}\right)$, of DFT solutions as(1c)$$\rho_{el}\left(\vec{r}\right)=\sum_{q}f_{q}{\left|\varphi_{q}\left(\vec{r}\right)\right|}^{2}$$
with the occupation probability given by the Fermi–Dirac distribution function with the electronic temperature $T_{el}$. This formula can be reformulated in terms of dipoles as(2a)$$\vec{P}=\frac{1}{V}\left(\sum_{j=1}^{N}\sum_{i=1}^{m}{\vec{d}}_{i}\right)$$
where the first sum runs over all atoms and the second runs over all bonds, represented as dipoles (the number of dipoles is set to *m*, and in the case of zinc blende and wurtzite, m=4). This representation ensures that polarization disappears in zinc blende but not in wurtzite. This representation is therefore fully compatible with the symmetry requirements.

It is necessary to discuss the problem of boundaries, or more precisely surface states, since only finite systems can exist. Boundaries include surface effects, i.e., charged surface states. The surface contribution is therefore defined as the difference between the crystal properties of the actual surface and the ideal continuation of the bulk. This can be illustrated as shown in Figure 4. So, we use Gibbs’ definition of a surface, where the bulk properties of a solid are homogeneous towards the surface, including symmetry, and the excess is associated with the surface. Therefore, the symmetry of the lattice is applied to the spontaneous polarization, and the surface properties are related to the surface polarity [36]. This removes the surface dependence and the associated spontaneous polarization ambiguity claimed in Refs. [13,14,15,16,17,18,19,20,21,22]. On the other hand, there is an experimental method for distinguishing between these two quantities. This is related to screening. In fact, in any macroscopic crystal, both insulating and conducting, band screening leads to a zero field in the bulk. This was identified by Meyer and Marx in their application of a slab model to the simulations of ZnO polar surfaces [39]. As they have shown, for macroscopically large samples, band bending due to the surface charge-related field leads to the appearance of the mobile screening charge in the conduction and valence bands, i.e., electrons and holes, respectively. The field associated with polar surfaces is screened, so the quantum wells in a system without polarization should have a zero field, regardless of the polarity of the surface. This zero-field result is confirmed in Figure 8a of Ref. [11], which shows the results of ab initio calculations of GaN/AlN MQWs with the gallium atoms in a perfect zinc blende AlN lattice, preserving zinc blende symmetry [11]. On the contrary, in the case of a spontaneously polarized medium such as wurtzite, the field in the well and the barrier appears as illustrated in Figure 8b of Ref. [11].

In order to determine the polarization based on the separation of the surface and bulk contributions, it is necessary to develop a model capable of obtaining the polarization of the bulk system without the surface. This could be carried out using Landau’s definition with the application of the single-cell system with proper periodic boundary conditions (PBCs). Figure 5 shows the two implementations of the Landau definition.

The first model, shown in Figure 5a, is constructed from the basic simulation of periodic cells by the controlled displacement of the cell along the 0z direction. The obtained dipole value of the AlN wurtzite unit cell is presented in Figure 6.

As can be seen, the dipole moment changes as a function of the location of the computational cell, i.e., with a shift along the c-axis. The moment undergoes a jump when the cell boundaries are crossed by Al or N atoms. In between, the dipole changes continuously, proving that no single dipole value could be associated with the cell. The calculation of the unique value of the polarization of AlN wurtzite is therefore not effective in this way, as any value in this range is equally valid. This is actually a demonstration of the influence of the reflection symmetry breaking with respect to the 0z-axis by the boundaries of the computational cell. In the correct approach, the geometry of the computational procedure (i.e., cell geometry) should be compatible with the reflection symmetry. Any symmetry breaking that introduces additional components, such as the cell boundaries, which will result in an incorrect polarization value.

A second model, shown in Figure 5b, is essentially an extension of the concept of bond formation by electron charge redistribution, as shown in Figure 3. The dipole moment of the cell is calculated as a sum of the moments obtained for each atom (*i*) separately [1]:(2b)$$\vec{d}=\sum_{i=1}^{m}{\vec{d}}_{i}$$
where the dipole related to atom (*i*) is calculated as(2c)$${\vec{d}}_{i}=\left[Z_{i}{\vec{R}}_{i}-\int_{V_{i}}\vec{r}\rho_{el}\left(\vec{r}\right)d^{3}r\right]e\;$$
where $V_{i}$ is the volume of the cell, centered on atom (*i*). Thus, any one atom is surrounded by the cell that is symmetric with respect to the inversion, i.e., to reflections relative to all three axes. The overall electron density field is a periodic repetition of a single computational cell. This new atom-associated cell is then defined as the computational cell of the volume $V_{i}$ centered on the selected atom (*i*). These atom-associated cells have their electron charge normalized to the valence charge of specified atoms. For atomic cell (*i*), the calculated dipole corresponds to the emergence of the moment due to the displacement of the atom (*i*) charge. In summary, in the region of the overlap of four cells, marked by different colors in Figure 5, the density is equal to the density obtained from DFT calculations. In the other cell, the same total density is obtained from the contribution from the marked cells and the neighboring repetition cells. They are marked by dashed colored outlines throughout the entire space. Therefore, the total electron density, which is composed of the sum of the contributions of all atoms, is equal to that obtained from ab initio calculations in the whole space. At the same time, this density is a patchwork of individual atomic contributions. The total dipole moment of the cell is the sum of the moment of all atoms. Since the total charge of single-atom cell (*i*) is electrically neutral, the dipole moment obtained does not depend on the coordinate system, so it could be added to the total. The total moment is the sum of the dipole moments of the cells, so the total moment divided by the cell volume gives the polarization of the solid. The geometry of the individual cells is symmetric with respect to the reflections relative to three axes, thus not breaking the reflection symmetry.

## 3. The Calculation Method

The majority of the ab initio calculations were performed using the commercial Vienna Ab initio Simulation Package (VASP) provided by the University of Vienna [40,41,42,43]. This density functional theory (DFT) code uses the momentum basis functional set to solve nonlinear Kohn–Sham equations. These planar wavefunctions are denoted by the momentum vector values $\vec{k}$. The maximum value of the momentum vector is determined by the energy cutoff, which is arbitrarily set using the maximum kinetic energy cutoff $E_{cut}=\frac{\hbar^{2}k^{2}}{2m}$. The density of the $\vec{k}$ points is determined by the size of the system $\left(L_{i},\;i=x,\;y,\;z\right)$ by the period boundary conditions (PBCs) $\left(k_{i}=\frac{2\pi }{L_{i}}\right)$. The same PBCs are applied to solve the coupled Poisson equation via Fourier series. In the present solution, the cutoff energy was set to $E_{cut}=400\;$eV.

The planar wavefunction set for the all-electron solution of the system consisting of the metal atoms boron, aluminum, gallium, and indium, and also nitrogen atoms, is prohibitively large, so that, even for a relatively small system size, the reduction in the basis is necessary. Therefore, the electron sets of all atoms are divided into two separate classes. The first set consists of the atomic core electrons. These electrons are not explicitly taken into account. In fact, this set consists of closed-shell electrons that are relatively unaffected by the crystal bonding. Therefore, the atomic cores are frozen, and only the polarization effects are taken into account. The second set, considered explicitly, consists of valence electrons and is equal to the total number of electrons in the computational cell. This separation requires a special formulation in which the Coulomb potential is replaced by the procedure using the regular function or even the set of matrix elements. In the VASP, Kresse’s standard for the norm-conserving or projector-augmented wave (PAW) potentials is available [44,45].

The standard ab initio approaches systematically underestimate semiconductor bandgaps, typically producing values around 30% smaller than those measured experimentally. Therefore, the standard DFT functional is supplemented by the Heyd–Scuseria–Ernzerhof (HSE) functional, which is essentially an extension of the standard DFT functional by the Hartree–Fock set of equations [46]. This implementation is numerically expensive but is optimal for the small size of the simulated systems. The experimental data are commonly used to verify the quality of the parameterization. The lattice parameters of the bulk wurtzite boron nitride, obtained from our ab initio calculations are $a_{BN}^{DFT}=2.542\;Å$ and $c_{BN}^{DFT}=4.202\;Å$. The synthesis of wurtzite BN is extremely difficult; nevertheless, the lattice parameters of w-BN were measured by X-rays, giving $a_{BN}^{exp}=2.550\;Å$ and $c_{BN}^{exp}=4.227\;Å$ [47]. Thus, for BN, the ab initio/X-ray agreement is reasonably good. DFT lattice data for wurtzite AlN are $a_{AlN}^{DFT}=3.113\;Å$ and $c_{AlN}^{DFT}=4.983\;Å$ [48]. They are in reasonable agreement with the X-ray measurement data of bulk AlN wurtzite: $a_{AlN}^{exp}=3.111\;Å$ and $c_{AlN}^{exp}=4.981\;Å$ [46]. The calculated values for wurtzite GaN are $a_{GaN}^{DFT}=3.196\;Å$ and $c_{GaN}^{DFT}=5.204\;Å$, remaining in good agreement with the X-ray data: $a_{GaN}^{exp}=3.1890\;Å$ and $c_{GaN}^{exp}=5.1864\;Å$ [47]. For InN, these data are $a_{InN}^{DFT}=3.571\;Å$ and $c_{InN}^{DFT}=5.742\;Å$ [49]. They are in good accordance with the experimental data for wurtzite InN: $a_{InN}^{exp}=3.5705\;Å$ and $c_{InN}^{exp}=5.703\;Å$ [50].

HSE approximation is able to obtain the energy bandgaps for wurtzite nitrides, which are in general agreement with the data from optical measurements. For wurtzite boron nitride, the obtained energy gap value is $E_{g}^{DFT}\left(BN\right)=6$.77 eV. The experimental data for wurtzite BN are scarce, and the measured bandgap is $E_{g}^{exp}\left(BN\right)=6.8\;\mathrm{eV}$, confirming good agreement with HSE approximation and the experimental results [51]. The HSE bandgap of AlN was $E_{g}^{DFT}\left(AlN\right)=6.19\;$eV, which is in good agreement with the experimental data of Silveira et al. ($E_{g}^{exp}\left(AlN\right)=6.09\;\mathrm{eV}$) [52]. The ab initio bandgap of w-GaN was $E_{g}^{DFT}\left(GaN\right)=3.41\;\mathrm{eV}$, which is in agreement with $E_{g}^{exp}\left(GaN\right)=3.47\;\mathrm{eV}$ [53,54]. The HSE bandgap of indium nitride was calculated to be $E_{g}^{DFT}\left(InN\right)=0.90\;\mathrm{eV}$. The optical InN bandgap was the subject of extensive debate, with the final constant set to $E_{g}^{exp}\left(InN\right)=0.65\;\mathrm{eV}$ [55,56,57].

The electron charge distribution is given as the set of the density values at the rectangular lattice points. Therefore, the data are actually a discrete representation of the continuous field. When calculating the dipole moment, the charge can first be summed over the plane perpendicular to the dipole vector. This generates the uniaxial density distribution shown in Figure 7. The number of density points may be controlled. The resulting density distribution is essentially symmetric, with no indication of a visible charge shift for the plot along the 0z direction. This confirms that the polarization is an extremely tiny effect, which can be determined by extremely precise calculations.

The electron density cannot be summed directly, as this introduces errors that are much larger than the calculated effect. Cubic spline functions were used to better mimic the smooth electron density. The plotted distribution is sufficient to integrate the density not only into the base cell but also into the shifted cell. As the atom-centered cell can be extended over neighboring cells, the density was calculated over three neighboring cells. Therefore, the spline approximation was made over the extended distance. The plots prove that the connection between cells is smooth, correctly recovering the periodic density distribution.

## 4. Results

### 4.1. Electric Dipole Calculations—Wurtzite

In the VASP, the electron density output is given as the values on the grid of the equidistant points parallel to all basic axes. In dipole calculation, the charges could be averaged (summed) over the plane perpendicular to the dipole axis. This gives the sequence of the charge distribution along the selected axis. Examples of such distributions are shown in Figure 7. The number of the subdivisions along the 0z-axis was changed so that the coarse-grained approximation to smooth charge distribution becomes more accurate as the number of subdivisions increases. In order to limit the computer resources required, the number of divisions along the 0x and 0y axes has not been changed and remains at 33. Therefore, the z-component of the dipole of the cell, and hence the polarization along the 0z-axis, depends on the number of divisions. Figure 8 shows the z-component of the polarization of the wurtzite nitrides BN, AlN, GaN, and InN plotted as a function of the number of divisions of the c lattice parameter.

In the case of wurtzite’s structure, the cell consists of four atoms: two N and two Me (B, Al, Ga, In) atoms. Our simulation of a BN cell employed the following parameters: $a_{BN}^{DFT}=2.542\;Å$ and $c_{AlN}^{DFT}=4.202\;Å$. The base area of the BN cell was $S_{BN}^{DFT}=5.631\;{Å}^{2}$, and the volume was $V_{BN}^{DFT}=2.351\;{Å}^{3}$. Our simulation of AlN employed a cell of the following geometry: $a_{AlN}^{DFT}=3.113\;Å$ and $c_{AlN}^{DFT}=4.982\;Å$; so, the base area was $S_{AlN}^{DFT}=8.382\;{Å}^{2}$, and the volume was $V_{AlN}^{DFT}=41.796\;{Å}^{3}$. In the case of GaN, these data were $a_{GaN}^{DFT}=3.196\;Å$ and $c_{GaN}^{DFT}=5.204\;Å$ and, accordingly, $S_{GaN}^{DFT}=8.843\;{Å}^{2}$ and $V_{GaN}^{DFT}=46.020\;{Å}^{3}$. Finally, the InN lattice parameters were $a_{InN}^{DFT}=3.571\;Å$ and $c_{GaN}^{DFT}=5.742\;Å$ and, accordingly, $S_{GaN}^{DFT}=11.041\;{Å}^{2}$ and $V_{GaN}^{DFT}=63.392\;{Å}^{3}$.

The simulation cell vectors for wurtzite were ${\vec{u}}_{1}=\left[a,0,0\right]$, ${\vec{u}}_{2}=\left[-a/2,a\sqrt{3}/2,0\right]$, and ${\vec{u}}_{3}=\left[0,0,c\right]$. So, the divisions only increase the number of points along the 0z-axis. The relatively small number of divisions is partially compensated for by summation over the plane perpendicular to the c-axis. However, it is possible that additional systematic errors were introduced.

The obtained dipole $P_{z}$ as a function of the number of intervals *N_3_* behaves similarly for all nitrides, and the magnitude of the dipole increases to reach the final asymptotic value. For GaN and InN, the dipole changes signs. Thus, the asymptotic behavior of the polarization is a clear confirmation of the nonzero-polarization value in all the nitrides. The fit to the data obtained gives the following approximate dependence (in $e/{Å}^{2}$):(3a)$${\vec{P}}_{z}\left(BN\right)=0.026-1.9/N_{3}\;$$(3b)$${\vec{P}}_{z}\left(AlN\right)=0.038-1.6/N_{3}\;$$(3c)$${\vec{P}}_{z}\left(GaN\right)=0.007-7.0/\left(N_{3}+29.8\right)$$(3d)$${\vec{P}}_{z}\left(InN\right)=0.013-4.3/\left(N_{3}+9.0\right)\;$$

Therefore, the obtained polarization values correspond to the asymptotic values for $N_{3}\to \infty$.

In addition to the z-component, the polarization values for the direction perpendicular to the 0z-axis could be obtained. The strict numerical constraints allow us to increase the number of points along a single axis, so the increase in point density is possible for the case of the 0y-axis. In the case of the 0x-axis, the increase for two principal axes is required. In this way, the polarization of the *y*-component of AlN was calculated, and the results are shown in Figure 9. As shown, the polarization component is much lower, ${\vec{P}}_{y}\left(AlN\right)\;≅2.7\times 10^{-4}\;e/{Å}^{2}$. It is not zero, but it is an extremely low value. Thus, the calculations give a small nonzero value, thus showing the accuracy of the representation of the density field. Any finite numbers of points cannot give the value of the polarization below certain limit, which in our case was $\Delta P\;\sim \;{\vec{P}}_{y}\left(AlN\right)\;≅2.7\times 10^{-4}\;e/{Å}^{2}$. In fact, this corresponds to the zero-polarization value, which is in accordance with the symmetry requirements.

### 4.2. Electric Dipole Calculations—Zinc Blende

Further verification of the basic model comes from the calculation of the zinc blende polarization values of these nitrides. According to the symmetry argument, the polarization is zero. The lattice constant of AlN was $a_{AlN-zb}^{DFT}=2.680\;Å$. The computational cell with a volume of $V_{AlN-zb}^{DFT}=31.421\;{Å}^{3}$ contains 6 atoms: 3 Al and 3 N. The computational cell vectors for zinc blende were ${\vec{u}}_{1}=\left[a,0,0\right]$, ${\vec{u}}_{2}=\left[-a/2,a\sqrt{3}/2,0\right]$, and ${\vec{u}}_{3}=\left[0,0,c\right]$. A straightforward calculation was therefore only possible for the z-component. The calculated result for the polarization of zinc blende AlN is shown in Figure 10. As can be seen, the polarization values are relatively high, but they decrease continuously. This is due to the fact that the grid density on the plane perpendicular to the triple axis is low. Therefore, some values for the perpendicular plane, compensating the bonds parallel to the triple axis, are missing. The error in the determination is therefore large. Nevertheless, these data are consistent with the zero value of spontaneous polarization in zinc blende crystals, in accordance with the symmetry arguments.

These data show the monotonous decrease in polarization values as the number of intervals increases. The following approximations have been made for these data:(4a)$${\vec{P}}_{z}\left(HSE\right)=0.0816+3.99/\left(13.85+N_{3}\right)\;$$(4b)$${\vec{P}}_{z}\left(PBE\right)=0.0808+3.92/\left(13.72+N_{3}\right)\;$$

The data prove that both approximations give essentially identical values of polarization. The difference is minor. On the other hand, the asymptotic value is not zero, which is related to the small number of points on the perpendicular plane, so the cancellation of the three dipoles at an angle, with that along the c-axis, is not complete. On the other hand, the data for wurtzite showed a monotonous increase in the dipole magnitude while the opposite was true for zinc blende. This again confirms the disappearance of the polarization in the latter case.

### 4.3. Spontaneous Polarization and Zero-Field Polarization (Berry Phase)

The above values were obtained for the zero electric field, as this is the only condition compatible with solving the Poisson equation using the Fast Fourier Transform (FFT) method. This is different from a spontaneous polarization state, which is defined as the appearance of the dipole moment and the electric field without any external contribution. The latter therefore assumes that the electric displacement field in the whole system is zero, i.e., $\vec{D}=0$ [32]. This assumption determines the relationship between the spontaneous polarization ${\vec{P}}_{i,s\;}={\vec{P}}_{o}$ and electric field ${\vec{E}}_{i,s\;}$ inside the polarized medium (the indices denote the following: *i*—internal, *s*—spontaneous) [3,32]:(5)$${\vec{P}}_{i,s}={\vec{P}}_{o}=-\varepsilon_{o}{\vec{E}}_{i,s}$$

The electric dipole vector is directed from a negative to a positive charge, whereas the electric field is the force acting on a positive charge; i.e., it is in the opposite direction. Suppose we consider an infinite polar slab. Then, the electric field associated with the spontaneous polarization outside the slab ${\vec{E}}_{e,s}$ (indices: e—external, i—internal) vanishes, i.e., ${\vec{E}}_{e,s}=\vec{D}=0$. In the calculation of spontaneous polarization employing Berry phase formulation [25,26], Resta assumed that the electric field vanishes, i.e., ${\vec{E}}_{i,B\;}=0$ (B—Berry state). From the spontaneous condition $\vec{D}=0$, it follows that Berry phase polarization ${\vec{P}}_{i,B\;}$ should vanish, i.e., ${\vec{P}}_{i,B}=0$. This is not the case; therefore, the condition $\vec{D}=0$ is not fulfilled in the Berry state, and thus, this state requires a nonzero external electric field, $\Delta {\vec{E}}_{iB}$, to be added to the spontaneous polarization field so that $\Delta {\vec{E}}_{iB}+{\vec{E}}_{i,s}=0\;$. This additional field obeys the linear regime with the continuity of the electric displacement field, ${\vec{D}}_{i}={\vec{D}}_{i}$; therefore, the external compensating field $\Delta {\vec{E}}_{eB}$. [3,32] is(6)$$\Delta {\vec{E}}_{eB}=\varepsilon \;\Delta {\vec{E}}_{iB}=\frac{{\vec{P}}_{o}\;\epsilon }{\varepsilon_{o}}$$
where ϵ is the dielectric permittivity. The application of this field induces the polarization change $\Delta \vec{P}$:(7)$$\Delta \vec{P}=\varepsilon_{o}\;\chi \;\Delta {\vec{E}}_{iB}=\chi {\vec{P}}_{o}$$
where the dielectric susceptibility is χ=ϵ−1. Since the Berry polarization ${\vec{P}}_{B}$ is the sum of the spontaneous polarization ${\vec{P}}_{o}$ and polarization change $\Delta \vec{P}$, i.e., ${\vec{P}}_{B}={\vec{P}}_{o}+\Delta \vec{P}$, we obtain the following final result:(8)$${\vec{P}}_{B}=\epsilon \;{\vec{P}}_{o}$$

Thus, Berry (zero-field) polarization differs from spontaneous polarization by a factor equal to the dielectric permittivity of the material. Therefore, the data from Berry (zero-field) polarization, determined above, can be used to determine spontaneous polarization.

### 4.4. Multiquantum Well (MQW)/Superlattice Calculations—Wurtzite

Additional verification of the polarization values can be obtained indirectly from ab initio calculations of polar GaN/AlN, InN/GaN, BN/AlN, and InN/AlN superlattices and be used as multiquantum wells (MQWs) in optoelectronic devices. These structures are very thin, so the polarization-induced electric fields are not screened, resulting in a quantum-confined Stark effect (QCSE) [8,9,10,11,12]. In most cases, the Fermi level in the bulk semiconductor is pinned by the same defect on both sides of the structure. Thus, the Fermi level position, and hence the potential difference, is approximately zero throughout the entire barrier–well system, so that the electric fields in the well $\left(E_{w}\right)$ and in the barrier $\left(E_{b}\right)$ are proportional only to the polarization difference [9,10,11,31]. Therefore, it is assumed that the potential is periodic with respect to a single barrier–well length, from which the electric field in the wells $E_{w}$ and in the barriers $E_{b}$ can be derived, as follows [9,10,11]:(9a)$$E_{w}=\frac{b\left(P_{w}-P_{b}\right)}{\varepsilon_{o}\left(w\varepsilon_{b}+b\varepsilon_{w}\right)}$$(9b)$$E_{b}=\frac{w\left(P_{b}-P_{w}\right)}{\varepsilon_{o}\left(w\varepsilon_{b}+b\varepsilon_{w}\right)}$$
where *w* and *b* are the thicknesses of the well and barrier, respectively, $\varepsilon_{w}$ and $\varepsilon_{b}$ are the dielectric constants of the well and barrier, and $\varepsilon_{o}$ is the permittivity of the vacuum. In these equations, it was assumed that the potential jumps [9,58], due to dipole layers at heterointerfaces, cancel out. These fields may be used to obtain the following polarization difference:(10)$$\Delta P=P_{w}-P_{b}=\frac{\;\varepsilon_{o}\left(w\varepsilon_{b}+b\varepsilon_{w}\right)E_{w}}{b}=-\frac{\;\varepsilon_{o}\left(w\varepsilon_{b}+b\varepsilon_{w}\right)E_{b}}{w}$$

Such wurtzite structures have been calculated using the ideal lattice positions of the BN, AlN, GaN, and InN lattices. The model has been created so that the metal atoms are at the sites of the single nitride semiconductor lattice, e.g., Ga atoms are in the AlN lattice. No relaxation was allowed, so the layers are lattice-strained. In this case, the lattice is a pure single wurtzite semiconductor. Since we used an identical number of both metal layers, the thicknesses of the well and the barrier are therefore identical in these simulations, i.e., w=b. From these results, the fields in the wells and barriers were obtained by a linear fit to the potential profiles, as shown in Figure 11.

In fact, the polarization obtained is strongly influenced by the piezoelectric effects. Thus, in the strained lattice, the z-component of the polarization is(11)$$P_{3}=P_{3,0}+\epsilon_{311}\varepsilon_{11}+\epsilon_{333}\varepsilon_{33}$$
where $P_{3,0}$ is the z-component of the spontaneous polarization ${\vec{P}}_{o}$; $\epsilon_{311}$ and $\epsilon_{333}$ are piezo constants; and $\varepsilon_{11}$ and $\varepsilon_{33}$ strain tensor components. The properties of the strained barrier–well systems obtained in these simulations are summarized in Table 1.

In this table, the strain component for the native lattice is zero, the second layer is assumed to be strained according to the lattice parameter difference, i.e., $\varepsilon_{\alpha \alpha }=\frac{a_{i,\alpha \alpha }-a_{j,\alpha \alpha }}{a_{j,\alpha \alpha }}\;$, where *i* and *j* denote the well and barrier (w and b), respectively, and α=1,3 (coordinates).

In summary, polarization in strained systems has two components: spontaneous and piezo. The piezo has two contributions, related to the enforced strain along the c-axis and on the perpendicular plane, i.e., the strain tensor components $\varepsilon_{11}$ and $\varepsilon_{33}$, respectively. These data are not sufficient to calculate both piezo constants as we have a single equation form only. Therefore, the second system was designed so that the strain on the plane is identical, i.e., lattice-compatible. The layers are therefore not strained along the c-axis, so $\varepsilon_{33}=0$. These systems are therefore subjected to planar strain. The ab initio calculated electric fields along the c-axis of these superlattices are shown in Figure 12.

These polarization values can be used to determine the piezo constants. The polarization difference in a lattice-strained system is(12)$$\Delta P_{l}=\Delta P_{0}+\epsilon_{311}\varepsilon_{11}+\epsilon_{333}\varepsilon_{33}$$
where $\Delta P_{0}$ is the spontaneous polarization difference in the well and the barrier, $\varepsilon_{11}$ and $\varepsilon_{33}$ are the strain components of the strained layer (the second layer is not strained, so the strain component is zero), and $\epsilon_{311}$ and $\epsilon_{333}$ are piezo constants of the strained layer. In the case of a plane-strained system, the strain z-component is zero; therefore, the polarization difference is(13)$$\Delta P_{p}=\Delta P_{0}+\epsilon_{311}\varepsilon_{11}$$

From this set of data, the first piezo component can be obtained as(14)$$\epsilon_{311}=\left(\Delta P_{p}-\Delta P_{0}\right)/\varepsilon_{11}$$
and the second as(15)$$\epsilon_{333}=\left(\Delta P_{l}-\Delta P_{0}-\Delta P_{p}\right)/\varepsilon_{33}$$

The data for the plane-strained superlattices are given in Table 2. Using the data from Table 1 and Table 2 and applying Equations (14) and (15), the piezo constants were obtained and are shown in Table 3.

In this table, the strain component for the native lattice is zero, the second layer is assumed to be strained on the plane perpendicular to the c-axis, according to the lattice parameter difference, i.e., $\varepsilon_{33}=\frac{c_{i,33}-c_{j,33}}{a_{j,33}}$, where *i* and *j* denote the well and barrier (w and b), respectively.

### 4.5. Multiquantum Well (MQW)/Superlattice Calculations—Zinc Blende

Similar calculations were carried out for the zinc blende superlattice of the nitrides. The PBE approximation was used to obtain elongated profiles. As shown previously, both HSE and PBE approximations give identical polarization values. The potential profiles are shown in Figure 13. These data indicate that the electric fields in both the GaN well and the AlN barrier are extremely small, confirming the absence of the polarization-induced fields in the zinc blende lattice.

From the linear approximation, the following fields were obtained: AlN/GaN − $E_{w}=2.151\times 10^{-5}\;V/Å$, $E_{b}=-1.897\times 10^{-4}\;V/Å$, GaN/InN − $E_{w}=3.515\times 10^{-4}\;V/Å$, and $E_{b}=8.687\times 10^{-4}\;V/Å$. It can therefore be concluded that these data indicate the absence of polarization-induced fields in this structure.

### 4.6. Critical Comparison of the Results

The dielectric permittivity of the nitrides can be determined using the Green function formulation, but this approximation is subjected to relatively high error. Therefore, experimental data were used: wz-BN: $\varepsilon_{wz-BN}=6.85$, wz-AlN: $\varepsilon_{wz-AlN}=10.31$, wz-GaN $\varepsilon_{wz-GaN}=10.28$, and wz-InN $\varepsilon_{wz-InN}=14.61$. These data were used to determine the polarization values listed in Table 1. For the comparison, the data obtained in Refs. [9,27,28] as compiled in Ref. [33] are listed.

These polarization values are in general agreement with those obtained in Refs. [11,27,29]. They are very different from those obtained in Ref. [30]. The piezo constants are of the same order, but the values are quite different. The piezo values in the present work could potentially be affected by additional charges on the heterostructures. Therefore, much greater efforts should be made in the future to derive reliable piezo parameters from wurtzite semiconductors. One of the most promising is the application of the model presented in this paper.

## 5. Summary and Conclusions

A new way of summarizing the results obtained in this work will be used, following the following basic scheme: (i) the state of the art before this publication, (ii) the results of the present work, and (iii) the state of the art after this publication.

The state of the art before this publication may be summarized as follows:

(a)The application of the Landau model to infinite solids is questioned, and the separation of surface and polarization effects is claimed to be impossible [14,15,16,17,18,19,20,21,22].(b)Spontaneous polarization as a bulk quantity has been redefined according to the arguments used in Berry phase formalism [25,26].(c)Berry phase calculations of the spontaneous polarization of the nitrides gave drastically different data [27,28,29,30].(d)The slab calculation gave data [11] that are, in principle, compatible with several Berry phase results [27,29] but not with the second set [12,30]. Nevertheless, the difference is considerable, which can also be attributed to the surface charge in the slab model [11]. Therefore, the slab results cannot be treated as final.

The results presented in this publication may be summarized in the following way:

(a)Superlattice calculations provide data on polarization difference that is in fundamental agreement with all the results, i.e., both the Berry phase and slab results [33].(b)Polarization as a bulk quantity of an infinite solid is redefined, and separation into spontaneous polarization and surface effects is proposed. This separation is confirmed by the experimental data from red shift presence and absence for wurtzite and zinc blende MQWs, respectively.(c)A geometric model allowing the calculation of spontaneous polarization as an electric dipole density is formulated (based on the Landau definition).(d)It is shown that some previously proposed models of the polarization give an incorrect picture of the phenomenon, mixing polarization and polar surface effects.(e)The spontaneous polarization of wurtzite nitrides is calculated and shows that the c-axis component *P_x_* is nonzero and that the others, *P_x_* and *P_y_*, are zero.(f)The calculated spontaneous polarization of zinc blende nitrides is zero.(g)The obtained polarization values, *P_z_*, of wurtzite nitrides are in general agreement with the Berry phase results of Bernardini et al. [27,30] and differ from those of Dreyer et al. [30].(h)The obtained polarization values, *P_z_*, of wurtzite nitrides are in agreement with those derived from superlattice calculations [33].

The state of the art after this publication can be described as follows:

(a)Spontaneous polarization of an infinite solid is defined as a bulk quantity.(b)Spontaneous polarization can be calculated using the Landau formulation with a geometric model.(c)Polar surfaces are different objects, independent of spontaneous polarization.(d)Spontaneous polarization of nitrides, having been further checked in this publication, shows basic agreement for all wurtzite nitrides.(e)Piezoelectric effects for wurtzite nitrides are correctly obtained.

In conclusion, it is stated that the present work has made considerable progress in the basic understanding of the spontaneous polarization of infinite solids as a fundamental property, as well as polar surfaces as different objects, and also in the determination of the parameter values of wurtzite and zinc blende nitride semiconductors. However, the values cannot be treated as definite and sufficiently accurate due to the deficiencies of our computational resources, which affected the precision of the data obtained. Future investigations should include a high point density in three dimensions, which would improve precision, especially that of piezoelectric data.

## Figures and Tables

**Figure 1 materials-18-01489-f001:**
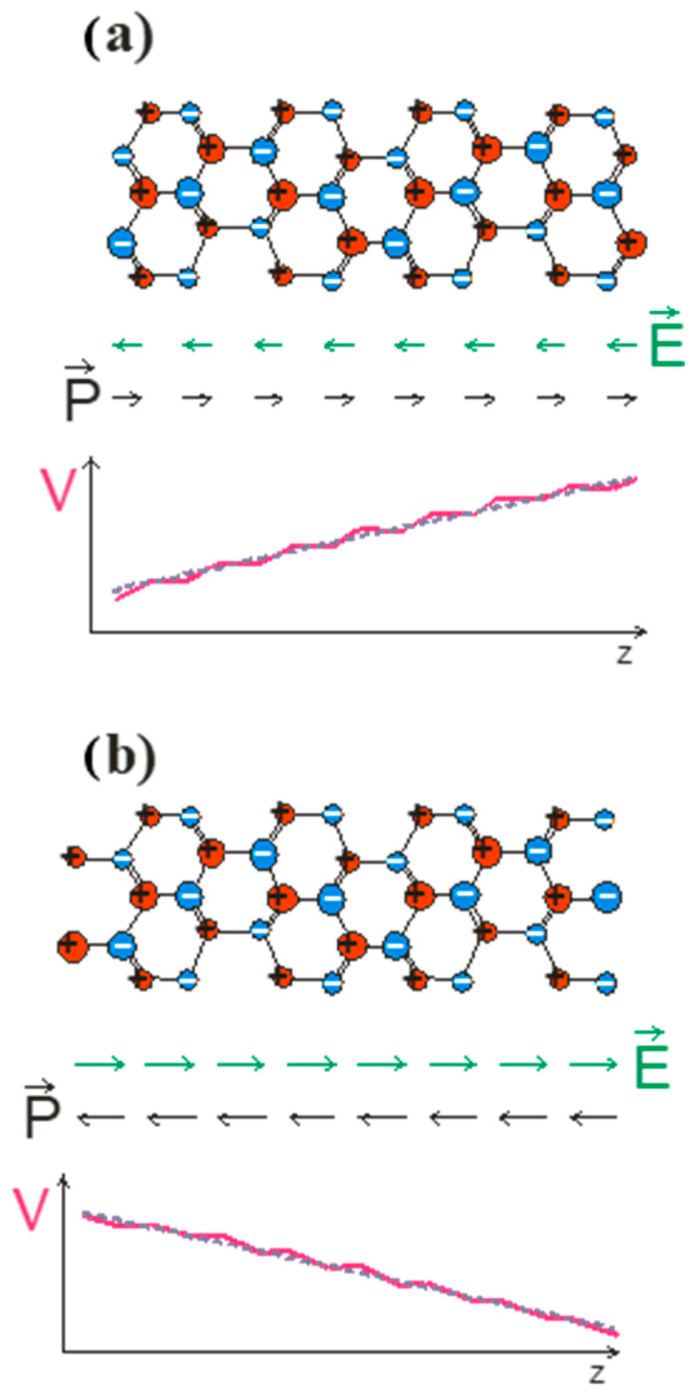
Zinc blende slabs of GaN with different termination: (**a**) by triple-bonded atoms and (**b**) by single-bonded atoms. Ga and N atoms are denoted by red and blue balls, respectively. The atoms located in second layer are denoted by smaller balls. In accordance with Ref. [22], it is assumed that polarization is induced by a charge shift from Ga to N atoms; therefore, Ga and N atoms are assumed to be positively and negatively charged. The green and black arrows represent the electric and polarization fields. The solid magenta and gray dashed lines represent layer- and slab-averaged electric potential profiles.

**Figure 2 materials-18-01489-f002:**
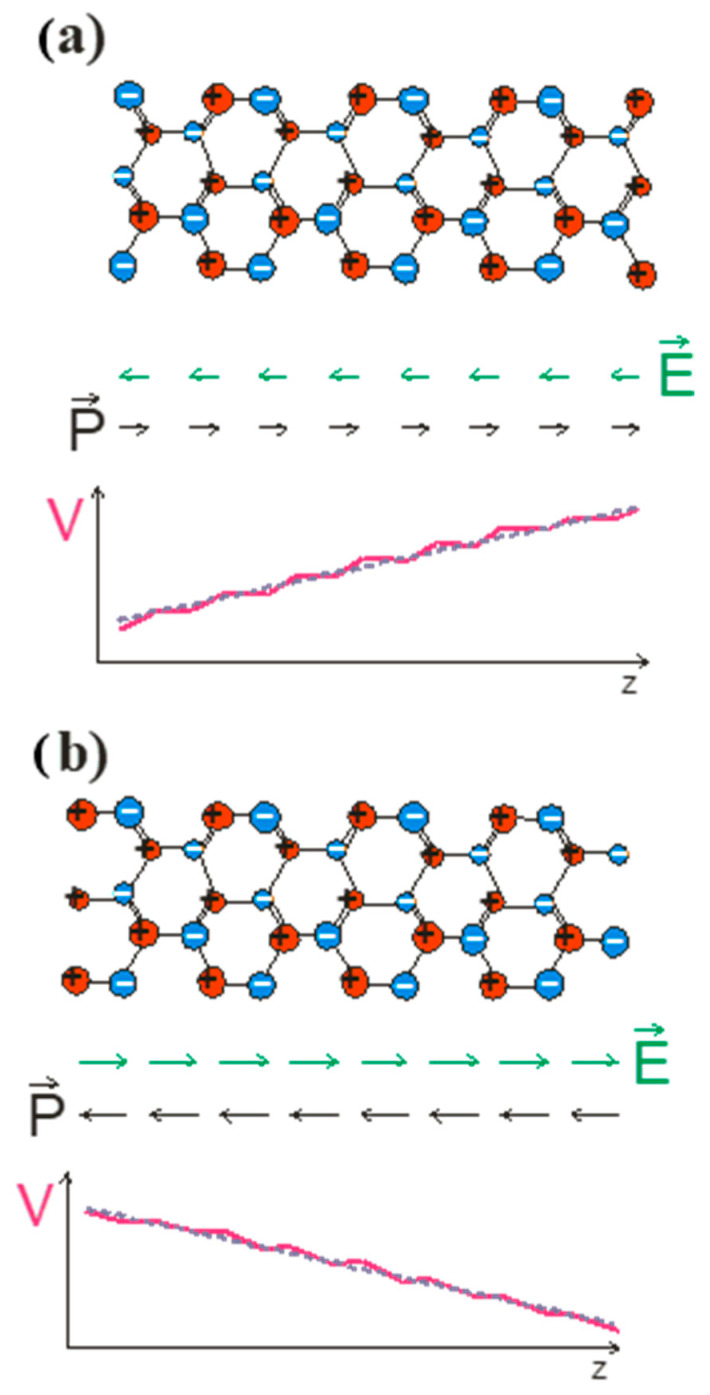
Wurtzite slabs of GaN with different termination: (**a**) by triple-bonded atoms and (**b**) by single-bonded atoms. Ga and N atoms are denoted by red and blue balls, respectively. The remaining symbols are also denoted as in Figure 1.

**Figure 3 materials-18-01489-f003:**
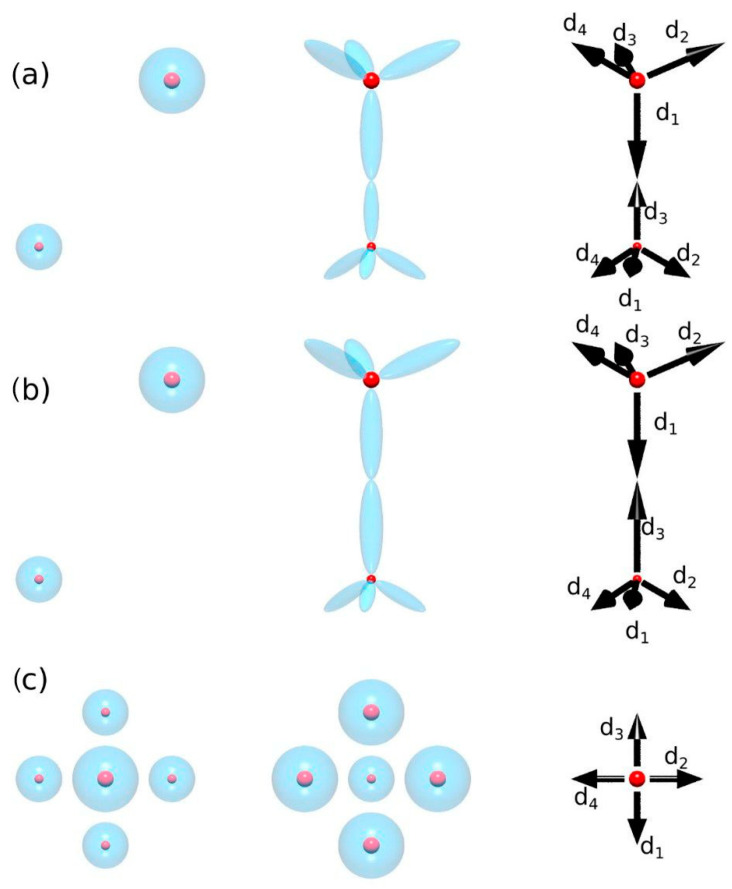
Emergence of polarization in the bonding of crystals: left—initial distribution of electron and protonic charge; center—distribution of the charge in the bonding; right—dipole representation of the charge in the bonding d1, …, d4. The diagrams present (**a**) zinc blende, (**b**) wurtzite, and (**c**) an ionic crystal. Blue and red denote electric and protonic charge, respectively.

**Figure 4 materials-18-01489-f004:**
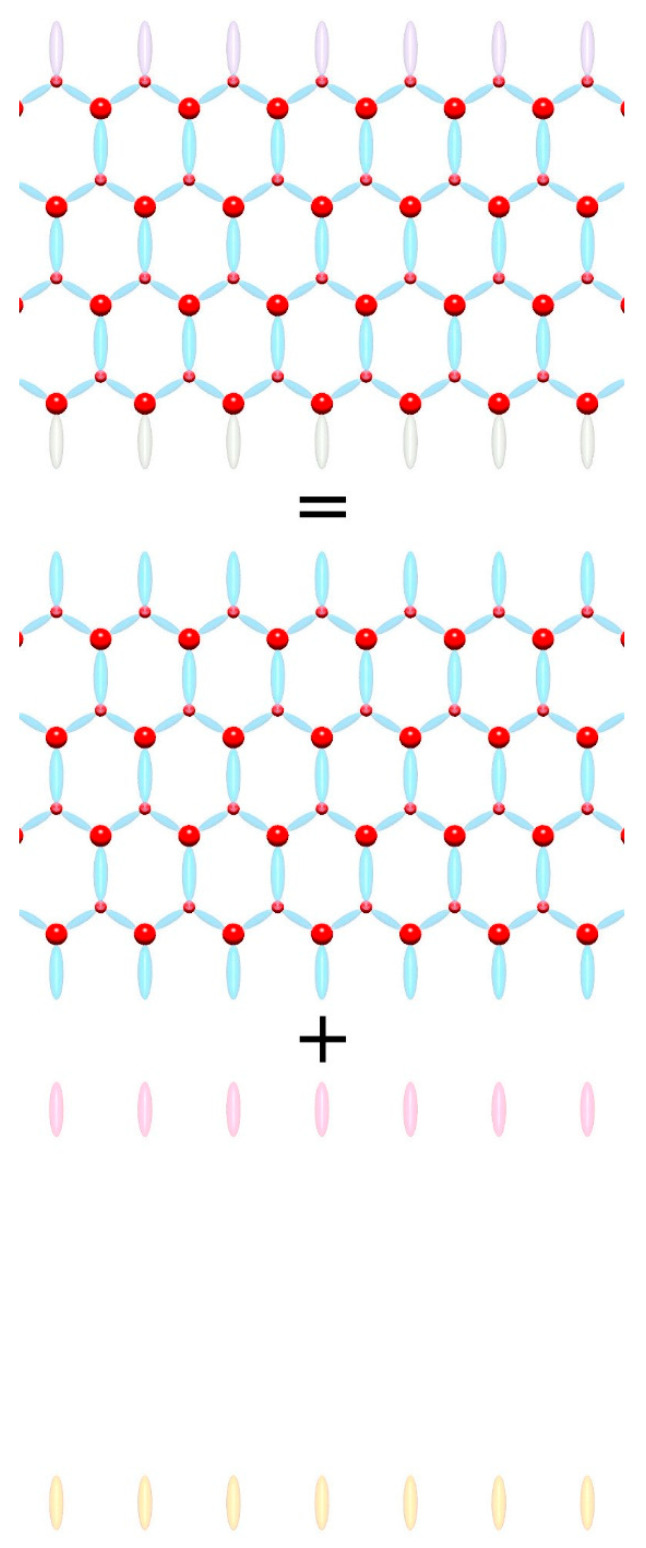
Separation of the polarization and the surface states. Top—the finite slab includes contributions from both polarization and the surface (surface quantum states) at both sides, different from the bulk and also possibly from each other; center—ideal polarization system; bottom—contribution from the surface states.

**Figure 5 materials-18-01489-f005:**
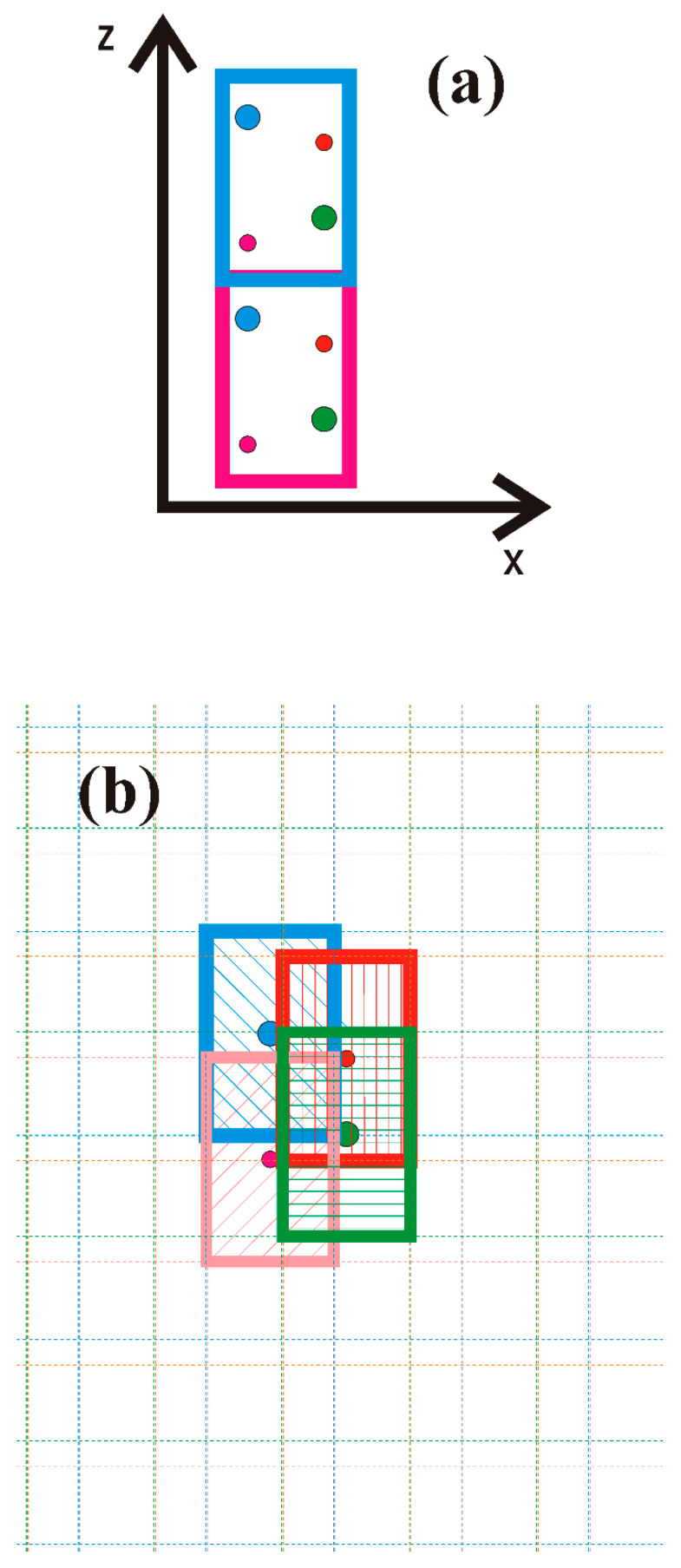
Two models used for calculation of the dipole of the unit cell of the nitride semiconductor: (**a**) standard cell—with a shift along the 0z-axis; (**b**) local atomic charge redistribution model—composed of the cells associated with the atoms. The colors denote 4 atoms: blue and green—nitrogen, red and magenta—metal (B, Al, Ga, In). The thick lines mark the cells used in the calculation of the dipole, and the dashed lines denote the multiple copies, spanning the entire space. The color of the lines denotes associations with the atoms.

**Figure 6 materials-18-01489-f006:**
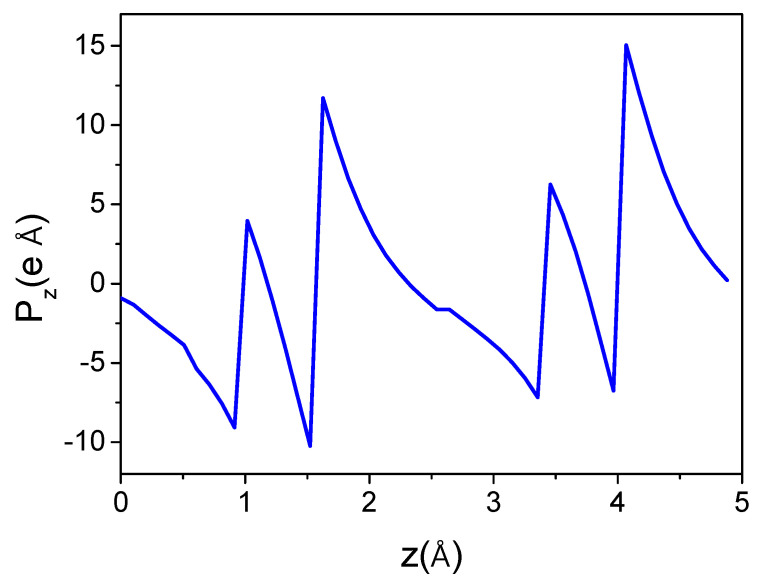
Dipole moment of the AlN wurtzite unit cell as a function of the shift of the cell along the c-axis.

**Figure 7 materials-18-01489-f007:**
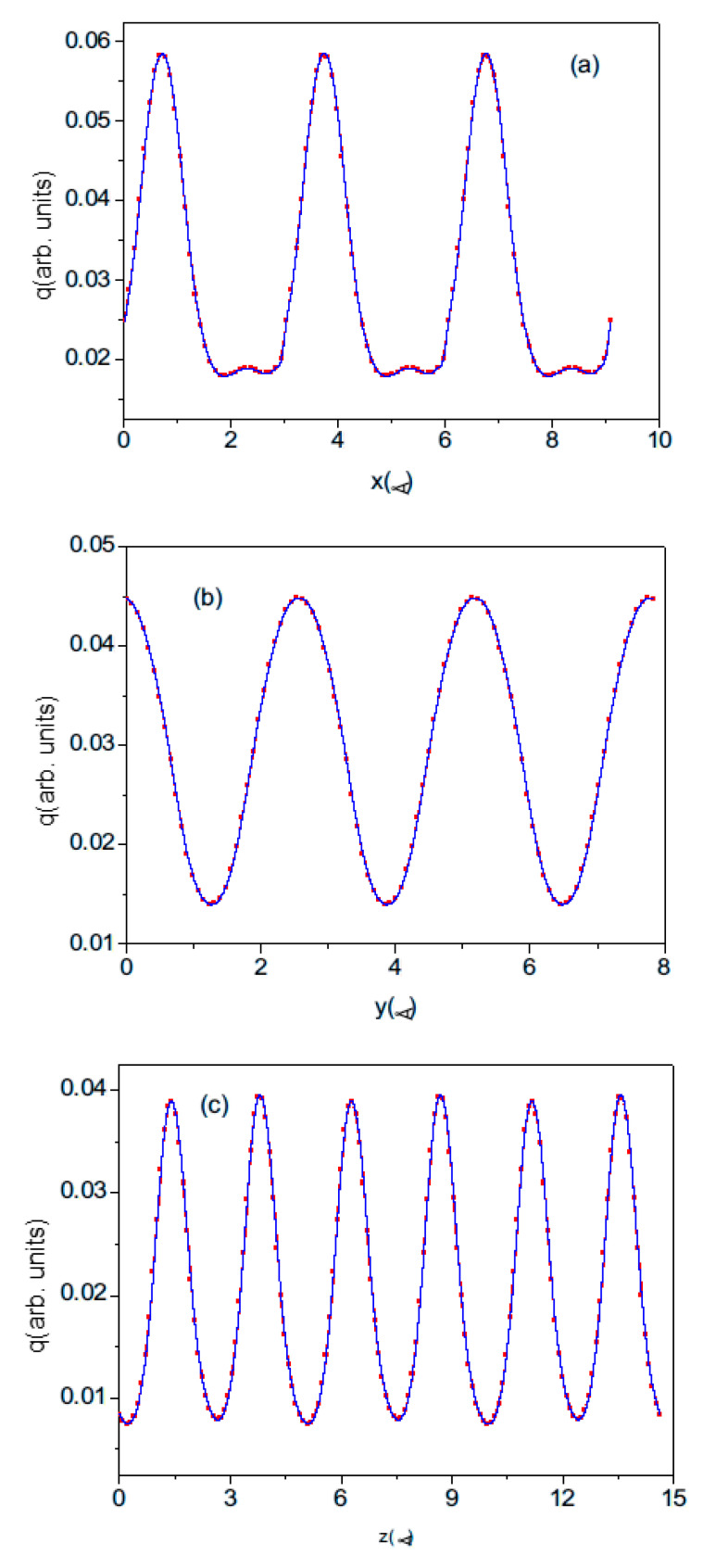
Axial density distribution for wurtzite AlN. The distribution is plotted over a length of 3 lattice constants along (**a**) the 0x-axis (over 9.338 Å), (**b**) the 0y-axis (over 8.087 Å), and (**c**) the 0z-axis (over 14.944 Å). The red points represent the plane-averaged values obtained using DFT, and the blue line is a cubic spline approximation of these data.

**Figure 8 materials-18-01489-f008:**
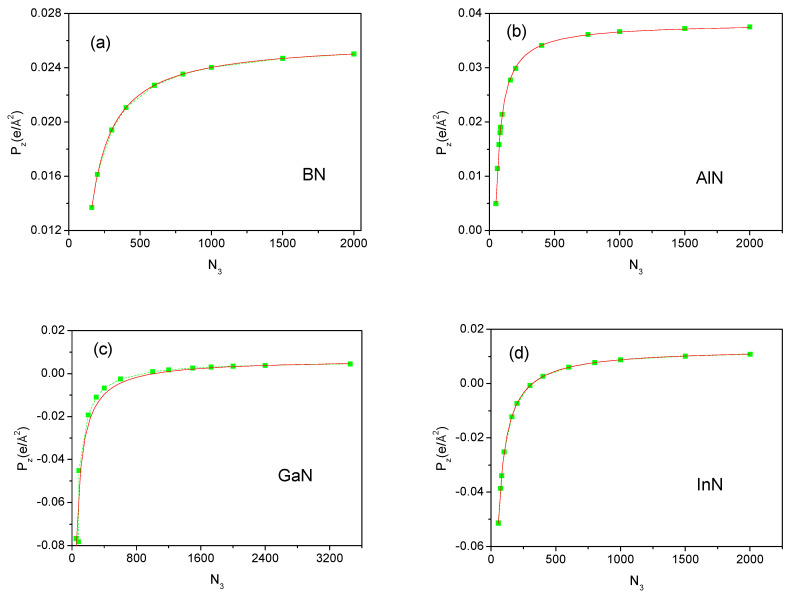
Spontaneous polarization z-component, $P_{z}$, of the wurtzite nitrides (**a**) BN, (**b**) AlN, (**c**) GaN, and (**d**) InN as a function of the number of divisions of the cell along the 0z-axis: *N*_3_. The number of divisions along the two other axes was $N_{1}=N_{2}=33$. The green squares are calculated values of polarization; green dashed lines are for guiding the eye only; the red line is an approximation in accordance with Equation (3a)–(3d).

**Figure 9 materials-18-01489-f009:**
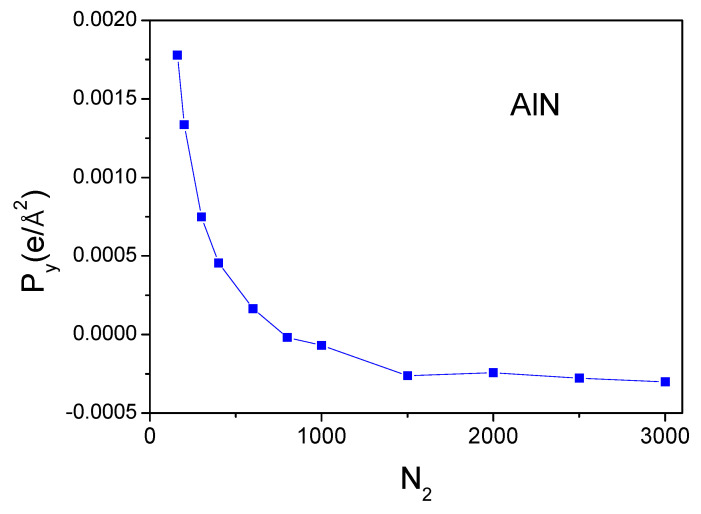
Spontaneous polarization y-component, $P_{y}$, of wurtzite AlN as a function of the number of divisions of the cell length along the 0y-axis: *N_2_*. The number of divisions along the two other axes was $N_{1}=33$ and $N_{3}=49$.

**Figure 10 materials-18-01489-f010:**
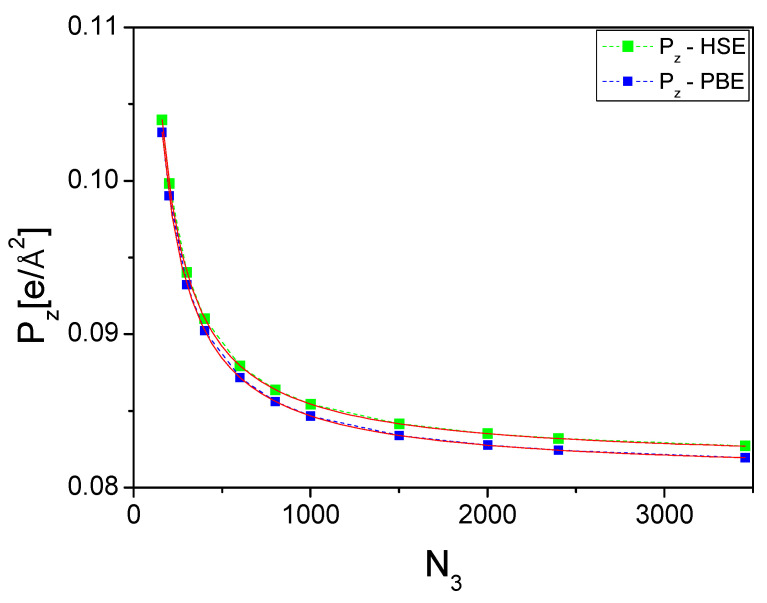
Spontaneous polarization z-component, $P_{z}$, of the zinc blende AlN as a function of the number of divisions of the cell length along the 0z-axis: *N_3_*. The number of divisions along the two other axes was $N_{1}=N_{2}=33$. The green and blue symbols denote data obtained for HSE and PBE approximations, respectively. The dashed lines are for guiding the eye, and the red solid lines are approximations in accordance with Equation (4a) and (4b).

**Figure 11 materials-18-01489-f011:**
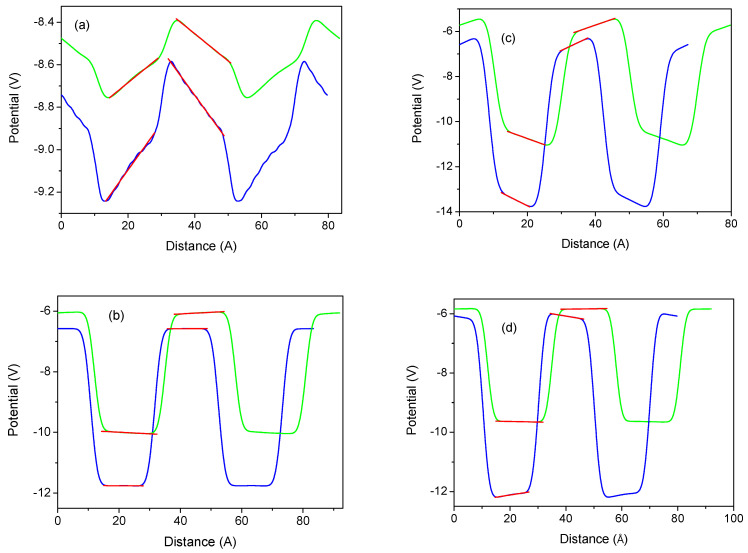
Averaged electric potential profiles along the 0z-axis in the wurtzite superlattice vs. the distance measured in the metal atomic layers (ALs) determined for structures with 8 ALs for both the well and the barrier thicknesses (i.e., b=w): (**a**) AlN/GaN; (**b**) GaN/InN; (**c**) BN/AlN; (**d**) AlN/InN. The green and blue lines correspond to larger and smaller lattice parameters (i.e., fractionally strained and compressed), respectively. Red lines represent linear slopes of the potential, i.e., electric fields.

**Figure 12 materials-18-01489-f012:**
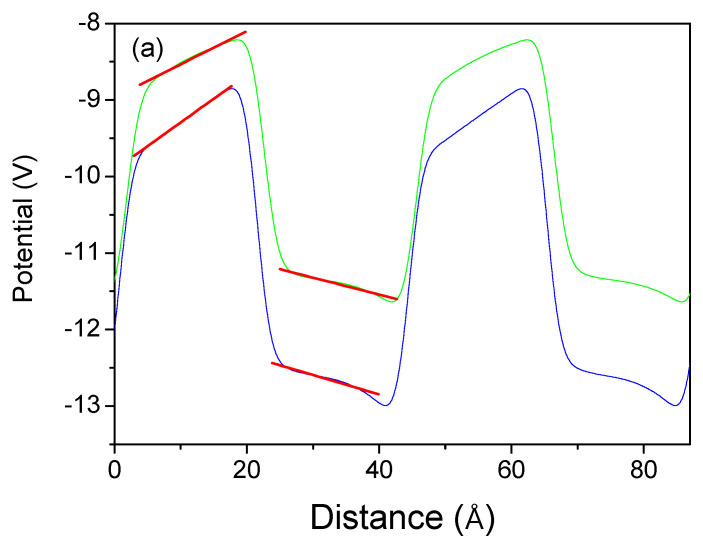
Averaged electric potential profiles along the 0z-axis in the wurtzite superlattice with a thickness of 8 metal atom layers (ALs) for both the well and the barrier: (**a**) GaN/InN; (**b**) BN/AlN. The system was strained on a plane, while it was relaxed along the c-axis. The green and blue lines correspond to plane-strained (i.e., b>w) and plane-compressed (i.e., b<w) lattices, respectively. The red lines represent linear slopes of the potential, i.e., electric fields.

**Figure 13 materials-18-01489-f013:**
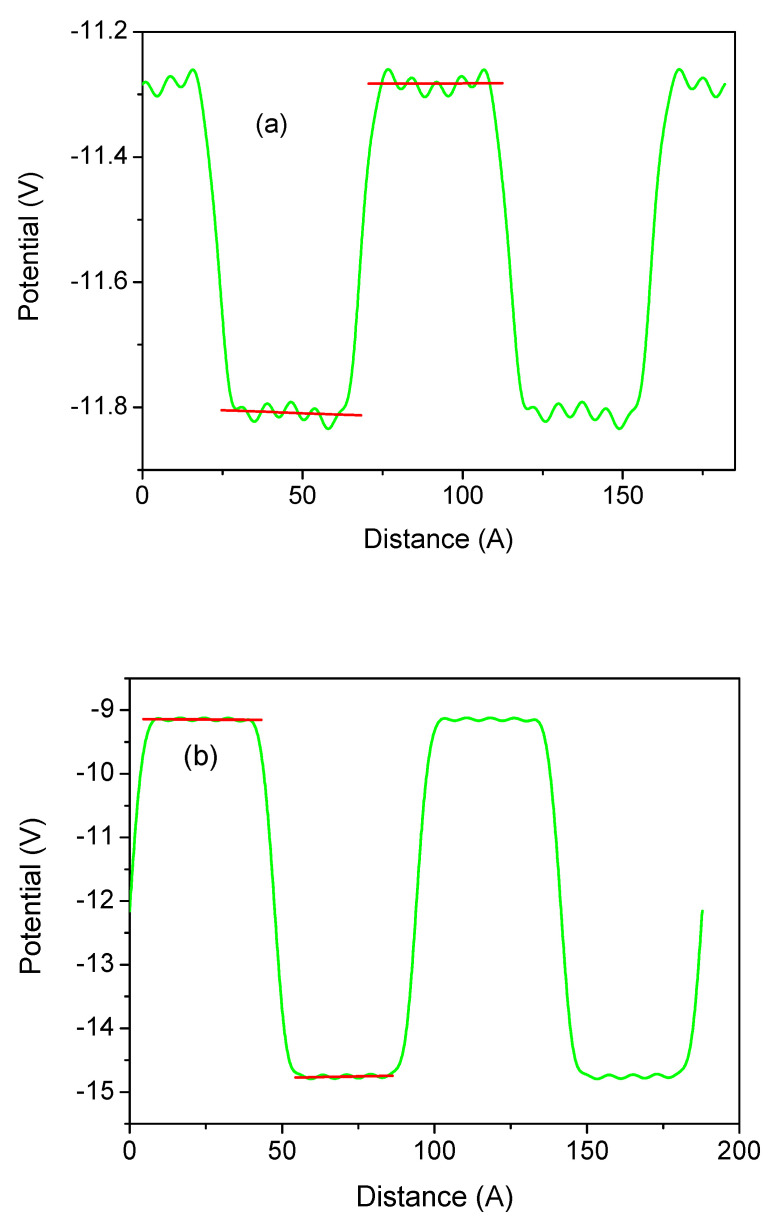
Averaged electric potential profiles along the [111] direction in the zinc blende superlattice: (**a**) AlN/GaN, (**b**) GaN/InN. The green and blue lines correspond to larger and smaller lattice parameters, respectively.

**Table 1 materials-18-01489-t001:** The properties of lattice-strained barrier–well systems.

System	Lattice	$\varepsilon_{11}\left(w\right)$ $\varepsilon_{33}\left(w\right)$	$\varepsilon_{11}\left(b\right)$ $\varepsilon_{11}\left(b\right)$	$E_{w}\left(V/Å\right)$	$E_{b}\left(V/Å\right)$	$\Delta P_{l}\left(C/m^{2}\right)$
AlN/GaN	AlN	−0.0259 −0.0428	0 0	0.0212	−0.0217	0.040
AlN/GaN	GaN	0 0	0.0266 0.0446	0.0129	−0.0128	0.023
GaN/InN	GaN	−0.1050 −0.9366	0 0	$-3.52\times 10^{-4}$	$4.64\times 10^{-4}$	$8.99\times 10^{-4}$
GaN/InN	InN	0 0	0.1173 0.1033	−0.00535	0.00526	0.012
BN/AlN	BN	−01834 0.1565	0 0	0.0706	−0.0759	0.081
BN/AlN	AlN	0 0	0.2246 0.1855	0.0529	−0.0523	0.111
AlN/InN	AlN	−0.2594 −0.1324	0 0	0.0150	−0.0158	$3.57\times 10^{-3}$
AlN/InN	InN	0 0	0.1282 0.1324	−0.00188	0.00136	0.040

**Table 2 materials-18-01489-t002:** The properties of the plane-strained (zero strain along the c-axis) barrier–well systems.

System	Strained	$\varepsilon_{11}\left(w\right)$ $\varepsilon_{33}\left(w\right)$	$\varepsilon_{11}\left(b\right)$ $\varepsilon_{33}\left(b\right)$	$E_{w}\left(V/Å\right)$	$E_{b}\left(V/Å\right)$	$\Delta P_{p}\left(C/m^{2}\right)$
GaN/InN	InN	−0.1050 0	0 0	0.0681	−0.0255	0.100
GaN/InN	GaN	0 0	0.1173 0	0.0434	−0.0221	0.0705
BN/AlN	AlN	−01834 0	0 0	−0.2741	0.3269	0.449
BN/AlN	BN	0 0	0.2246 0	−0.2384	0.2595	0.375

**Table 3 materials-18-01489-t003:** Polarization (in $C/m^{2}$) and piezoelectric constants of the nitrides.

Property	Ref.	BN	AlN	GaN	InN
Spontaneous polarization, $P_{3}$	This work	0.061	0.059	0.011	0.014
	[29]		0.081	0.029	0.032
	[29]		0.09	0.034	0.042
	[30]		1.351	1.312	1.026
	[11]		0.09	0.019	0.028
Piezo constant, *ϵ*311	This work	−1.17	−0.99	−0.64	−0.83
	[27]		−0.60	−0.49	−0.57
	[29]		−0.53	−0.34	−0.41
	[30]		−0.676	−0.551	−0.604
Piezo constant, $\varepsilon_{333}$	This work	1.88	1.18	0.74	0.96
	[27]		1.46	0.73	0.97
	[29]		1.5	0.67	0.81
	[30]		1.569	1.02	1.328

## Data Availability

The raw data supporting the conclusions of this article will be made available by the authors on request.

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
