# Peer review of "Spontaneous and Piezo Polarization Versus Polar Surfaces: Fundamentals and Ab Initio Calculations"

_materials, 2025, doi:10.3390/ma18071489_

Round 1
Reviewer 1 Report
Comments and Suggestions for Authors
This paper addresses the critical issue of defining spontaneous polarization in infinite solids. The authors validate their model using ab initio calculations on wurtzite and zinc blende III-nitrides, comparing results with Berry phase and superlattice methods. However, revisions to improve the quality are essential for broader acceptance.
- Clarify why HSE was used for spontaneous polarization calculations but PBE for zinc blende superlattices.
- The piezoelectric parameters in Table 3 show significant variation from literature. Please clarify this.
- The paper mentions computational constraints but does not quantify their impact. A discussion on how finer grids or larger supercells might alter results would strengthen conclusions.
- The Figures should be presented by a more clear format and correct units (like A should be corrected).
Author Response
Reviewer #1
Referee criticism
Clarify why HSE was used for spontaneous polarization calculations but PBE for zinc blende superlattices.
Our response
This is our error in writing. In fact HSE was calculated at final stage. We corrected the manuscript, i.e. abstract. In Fig. 10 for ZB we have PBE and HSE data.
_____________
Referee criticism
The piezoelectric parameters in Table 3 show significant variation from literature. Please clarify this.
Our response
In fact variation of these parameters may be related to the fact that we are limited to dense grid in one directions and lower density grid in other. But this is the limit of our computational possibilities.
_____________
Referee criticism
The paper mentions computational constraints but does not quantify their impact. A discussion on how finer grids or larger supercells might alter results would strengthen conclusions.
Our response
Generally this error is relatively small for spontaneous polarization as these are separated components. As mentioned above in the case of coupling between different piezo components via Poisson effect this may be larger because in the perpendicular the number of points is small.
_____________
Referee criticism
The Figures should be presented by a more clear format and correct units (like A should be corrected).
Our response
We apologize for that. It is corrected in the new version. The Figures were also improved.

Reviewer 2 Report
Comments and Suggestions for Authors
The authors present the Polarization spontaneous/piezo and polar surfaces: fundamentals and their implementation in ab initio calculations. The research idea is well conceived. However, the research design needs some attention, and the manuscript needs some polish before publication. Though I have given more than eight comments, all the comments are minor. I would like to accept the paper after minor revision of the following remarks.
1- The overall English is okay, but some sentences sometimes have structural issues. I want the authors to double-check for typos and structural English mistakes.
2- It would be better for the authors and the readers if the manuscript follows the proper MDPI materials' manuscript template to divide the paper into sections and subsections. The present layout is a bit confusing.
3- Some figures are disbursed on multiple pages. Always align them on a single page.
4- Figure 4 colours are very dull, which tends to miss some information. I recommend using dark and bright colours for better visibility. Also, it would be difficult to understand Figure 4 in the grey version of the manuscript.
5- Provide the novelty statement in the abstract and the last introduction paragraph.
6- Provide the bibliographic reference of the underlying base equations.
7- Figure 13 is not visible because of bad colouring choice.
8- The conclusion should be changed and written as per MDPI guidelines. Also, provide the limitations of this study and future directions to the conclusion section.
Comments on the Quality of English LanguageYes, minor issues
Author Response
Reviewer #2
1- The overall English is okay, but some sentences sometimes have structural issues. I want the authors to double-check for typos and structural English mistakes.
Our response
We apologize for that. We corrected some sentences and carried out the language check.
_____________
Referee criticism
2- It would be better for the authors and the readers if the manuscript follows the proper MDPI materials' manuscript template to divide the paper into sections and subsections. The present layout is a bit confusing.
Our response
This paper contains important section describing new model. Therefore it is slightly different. As it is crucial we prefer to keep it that way. There are not other differences.
_____________
Referee criticism
3- Some figures are disbursed on multiple pages. Always align them on a single page.
Our response
We apologize for that. We corrected them accordingly.
_____________
Referee criticism
4- Figure 4 colours are very dull, which tends to miss some information. I recommend using dark and bright colours for better visibility. Also, it would be difficult to understand Figure 4 in the grey version of the manuscript.
Our response
We apologize for that. We corrected it accordingly.
_____________
Referee criticism
5- Provide the novelty statement in the abstract and the last introduction paragraph.
Our response
The summary and conclusion Section (5) is actually great novelty statement because it presents
- The state of art before publication
- The results in the paper
- The state of art after publication.
From these points it is clear what is the contribution of the manuscript and how important that is. We suggest that this form should be applied in any published paper. That would avoid redundant papers.
_____________
Referee criticism
6- Provide the bibliographic reference of the underlying base equations.
Our response
Some references are added.
_____________
Referee criticism
7- Figure 13 is not visible because of bad colouring choice.
Our response
We apologize for that. We corrected it accordingly.
_____________
Referee criticism
8- The conclusion should be changed and written as per MDPI guidelines. Also, provide the limitations of this study and future directions to the conclusion section.
Our response
The conclusions are written in new style. This is to prove that the paper (i) provides new results, (ii) they change the general scientific picture of the field. The limitations and future directions are added to conclusion Section.
Finally we would like to thank all Referees for their extensive corrections that has helped us to improve the manuscript considerably.

Reviewer 3 Report
Comments and Suggestions for Authors
This manuscript employs an original approach to address the polarization spontaneous/piezo and polar surfaces: fundamentals and their implementation in ab initio calculations. The authors also account for and develop the local model of spontaneous polarization and employ it to calculate spontaneous polarization. Besides the credit of developing an excellent and convincing model, much needed for the wurtzite nitride superlattice compounds, the authors put their model and comprehensive results in the right perspective and context, both method-wise and in relation to the class of materials. The discussion and the study as presented invokes notable insights into the realm of electronic/optical properties of group IIIA. The work presents new developments and brings new knowledge to the field. Conclusions are motivated and easy to perceive.
There are some minor queries/suggestions related to this excellent manuscript that need to be addressed before its publication, i.e., I recommend it for publication after a minor revision:
1: Title: The phrase “polarization spontaneous/piezo and polar surfaces” is not optimal -both grammatically and logically. It is not stylistically appropriate for a title. Besides, the title could be shortened.
2: The DFT level of theory (basically, the HSE exchange-correlation functional) should be mentioned in the abstract
3: Lattice parameters/bond lengths obtained in DFT calculations are usually considered meaningful up till the third digit after the decimal point. It is not necessary (and usually considered not credible to operate, state, and compare results that include the forth digit after the decimal point).
4: Fig. 11: graph lines are too thin for comfortable reading.
5: Have the authors tested the impact of different unit cell size(s) on the accuracy of the calculation results?
6: The authors have already mentioned some studies of the electronic/optical properties of group IIIA nitrides at DFT level of theory but fail to comment/include some very recent comparative studies of group IIIA nitrides at ab initio levels which also extend to mesoscopic scale using the phase field model. (e.g., MAM Filho, et al., Crystal Growth & Design 24 (11), 4717-4727 (2024), and also RR Pela, et al. Physical Chemistry Chemical Physics 26 (9), 7504-7514 (2024)) which should be acknowledged in the Introduction.
7: Some long sentences, misspellings, etc., still are noticeable throughout the text.
Author Response
Reviewer #3
Referee criticism
1: Title: The phrase “polarization spontaneous/piezo and polar surfaces” is not optimal -both grammatically and logically. It is not stylistically appropriate for a title. Besides, the title could be shortened.
Our response
The title was changed. It is shorter now
_____________
Referee criticism
2: The DFT level of theory (basically, the HSE exchange-correlation functional) should be mentioned in the abstract
Our response
Thank you for the comment. It is added now.
_____________
Referee criticism
3: Lattice parameters/bond lengths obtained in DFT calculations are usually considered meaningful up till the third digit after the decimal point. It is not necessary (and usually considered not credible to operate, state, and compare results that include the forth digit after the decimal point).
Our response
The custom in the x-ray is to use four digits. In the case of ab initio the precision is lower. We have kept four digits in the experimental data and three in ab initio.
_____________
Referee criticism
4: Fig. 11: graph lines are too thin for comfortable reading.
Our response
We apologize for the error. It is corrected now.
_____________
Referee criticism
5: Have the authors tested the impact of different unit cell size(s) on the accuracy of the calculation results?
Our response
No, we have used smallest single size calculation cell. Larger cells would cause more serious problems with the computational limitations.
_____________
Referee criticism
6: The authors have already mentioned some studies of the electronic/optical properties of group IIIA nitrides at DFT level of theory but fail to comment/include some very recent comparative studies of group IIIA nitrides at ab initio levels which also extend to mesoscopic scale using the phase field model. (e.g., MAM Filho, et al., Crystal Growth & Design 24 (11), 4717-4727 (2024), and also RR Pela, et al. Physical Chemistry Chemical Physics 26 (9), 7504-7514 (2024)) which should be acknowledged in the Introduction.
Our response
We thank the Referee for these recent publications. These are excellent papers. But they consider nanorods which are much above the calculation sizes used in our paper. So we cannot refer them in the context of our results.
_____________
Referee criticism
7: Some long sentences, misspellings, etc., still are noticeable throughout the text.
Our response
We apologize for that we corrected the manuscript linguistically. The changes are marked by blue color.
_____________

Round 2
Reviewer 1 Report
Comments and Suggestions for Authors
All my concerns have been addressed, and the present form can be considered published.